# Critical role of slags in pitting corrosion of additively manufactured stainless steel in simulated seawater

Shohini Sen-Britain [1], Seongkoo Cho [1], ShinYoung Kang[1], Zhen Qi [1], Saad Khairallah[1], Debra Rosas[1], Vanna Som[1], Tian T. Li [1], S. Roger Qiu[1], Y. Morris Wang [2], Brandon C. Wood [1] & Thomas Voisin [1] ✉

Pitting corrosion in seawater is one of the most difficult forms of corrosion to identify and control. A workhorse material for marine applications, 316L stainless steel (316L SS) is known to balance resistance to pitting with good mechanical properties. The advent of additive manufacturing (AM), particularly laser powder bed fusion (LPBF), has prompted numerous microstructural and mechanical investigations of LPBF 316L SS; however, the origins of pitting corrosion on as-built surfaces is unknown, despite their utmost importance for certification of LPBF 316L SS prior to fielding. Here, we show that Mn-rich silicate slags are responsible for pitting of the as-built LPBF material in sodium chloride due to their introduction of deleterious defects such as cracks or surface oxide heterogeneities. In addition, we explain how slags are formed in the liquid metal and deposited at the as-built surfaces using high-fidelity melt pool simulations. Our work uncovers how LPBF changes surface oxides due to rapid solidification and high-temperature oxidation, leading to fundamentally different pitting corrosion mechanisms.

At the beginning of the twenty-first century, the direct cost of corrosion in the United States was estimated to be approximately $276 billion per year, which represented ≈3 percent of the national gross domestic product[1]. Factoring in the indirect cost, this estimation adds up to a minimum of $550 billion. The nature and cause of corrosion can vary drastically by sectors, applications, materials, or environments. In transportation industries, including aerospace, naval, and terrestrial, aqueous corrosion has a major impact on the cost of maintaining, repairing, or replacing metallic components. Of the many types of aqueous corrosion, pitting corrosion in chloride solutions is one of the most difficult to identify or predict due to its small length scale and stochastic behavior. The first step in pitting corrosion is the local breakdown of a protective surface oxide. The exposed underlying metal is then dissolved, and the process eventually transitions to a self-sustaining reaction causing the stable growth of pits within the occluded environment. These pits are typically deep craters on the order of 10–100 μm that can result in stress localization and initiate catastrophic failure.

316L stainless steel (316L SS) is the material of choice for structural applications that require both high strength and good resistance to pitting corrosion[2,3]. Due to concerns with energy consumption, cost, and environmental impacts of manufacturing, fabricating components with reduced weight while minimizing waste is crucial. As such, additive manufacturing (AM) techniques are taking the front row in producing near net-shape parts with previously impossible geometries that transcend conventional manufacturing limitations. Laser powder bed fusion (LPBF), also called selective laser melting (SLM) or 3D printing, has become one of the most widely used AM techniques for its optimized ratio between maximum processable part size and minimal spatial resolution. Using a laser, layers of pre-alloyed powders are locally melted following a predefined pattern, one after another, until a full component is built. In this way, LPBF allows for building

[1]Lawrence Livermore National Laboratory, Livermore, CA, USA. [2]Department of Materials Science and Engineering, University of California Los Angeles, Los Angeles, CA, USA. ✉e-mail: voisin2@llnl.gov

parts of very complex geometries such as lattices or closed shells[4-7]. Most of the work related to LPBF SS316L has focused on understanding the relationship between processing, microstructures, and mechanical properties[3]. For instance, it has been shown that LPBF SS316L parts have much higher yield strength than conventional counterparts[8-16] while maintaining a reasonable ductility, which is a very encouraging result that opens up a door to more demanding applications. However, understanding, controlling, and potentially improving the resistance to pitting corrosion in chloride solution is still needed for validation and certification prior to fielding[3].

The microstructure of as-built LPBF 316L SS is the result of complex thermomechanical cycling during manufacturing, including local rapid melting and solidification and residual stresses. This leads to the formation and retention of so-called non-equilibrium rapid solidification cellular structures in the as-built parts. These structures consist of dendrites elongated along the [001] parent grain crystallographic orientation with interdendritic regions containing high densities of dislocations, trapped elements, and nano-precipitates[8]. Note that one of the elements trapped at cell walls is chromium, which is a critical participant in the formation of a uniform protective passivating oxide layer at the surface of 316L SS. These rapid solidification structures are found everywhere in the material, including near the surfaces. However, as-built surfaces are known to be more complex due to the presence of secondary phases[17,18].

It has been shown that post-processed (machined or polished) LPBF 316L SS offers a significantly higher resistance to pitting than the conventional counterpart[19-24], based on electrochemical polarization testing in NaCl solutions. This improvement has been attributed to an enhanced passivity[21,25] and the absence of manganese sulfides (MnS). MnS inclusions deplete local Cr concentrations or dissolve faster than the protective surface oxide, leaving regions of conventional 316L SS without protection against the chloride solution to become pit nucleation sites[26]. Notably, MnS inclusions do not form during LPBF due to elevated cooling rates[27,28]. Although this also applies to as-built surfaces, they often exhibit slightly lower pitting potential than post processed surfaces[21-23]. The origin of pitting on as-built surfaces of the LPBF 316L SS material needs to be discovered and the associated limitations understood before it can be certified. Most published works have tested only machined or finely polished surfaces[23,29,30]; however, this may be less relevant to complex industrial parts, for which most surfaces remain as-built.

In this work, we focus on uncovering the origin of pitting corrosion of LPBF 316L SS as-built surfaces and establishing a correlation to microstructure and processing parameters. We use transmission electron microscopy (TEM) to unveil the complexity of the surface oxides before and after corrosion testing under polarization currents or at open circuit potential (OCP). In particular, we show that as-built surfaces are covered by a highly protective Mn-rich silicon oxides nanolayer but that several microns slags of similar composition located only at as-built surfaces are the cause of pitting corrosion in LPBF 316L SS. We use high-fidelity 3D melt flow simulations to explain why these slags are found at the as-built surfaces.

## Results and discussion

Due to the addition of deoxidizer elements to 316L steels such as Si, Al, Mn, Mg, and Ca, a secondary phase known as slags or silicate islands form during solidification of conventional steels[31-33]. These slags are known to be detrimental, and their migration to the surface allows them to be machined away easily. This is because the amorphous slags offer little bonding strength to the underlying crystalline steel. For this reason, they can also act as pitting initiation sites[34,35].

In LPBF 316L SS, slag formation correlates with the laser path. For example, Fig. 1a shows top and side as-built surfaces. On the top surface, where laser tracks are clearly visible, slags form a continuous stripe along laser tracks. This is indicated by the presence of Mn, Si,

and O[23,29,36], as shown by SEM/EDS elemental maps in Fig. 1b and Supplementary Fig. 1a. On the side surfaces, slags are also present but in a different size, shape, and distribution (Fig. 1c and Supplementary Fig. 1b). In fact, they are much shorter, sometimes equiaxed, and randomly distributed without apparent correlation with laser tracks or surface roughness. In addition, particles are seen at the top surface as a result of spatter redeposition[37-40]. Note that no MnS inclusions, such as found in conventional 316L, are observed on the LPBF material.

After potentiodynamic testing in 0.6 M NaCl, we found pits on the top surface systematically nucleated at the slags between the laser tracks (Fig. 2a). On the side surface, where the secondary phase is randomly spread, the pits are randomly distributed. This shows that pit initiation sites are associated with the slags. Figure 2b, c and Supplementary Fig. 1c, d show scanning electron microscopy energy dispersive X-ray spectroscopy (SEM/EDS) measurements conducted on pitted surfaces. Interestingly, the data reveal that slags mostly survive pitting on both top (Fig. 2b) and side (Fig. 2c) surfaces. This contrasts with the effect of MnS inclusions in conventional 316L SS (absent in LPBF) that fully dissolve[26].

It is important to verify that pits also nucleate at slags under more practical conditions without an applied voltage bias. Polarization curves (Supplementary Fig. 2a) for both top and side surfaces, and the corresponding average $E_{Pit} - E_{OCP}$ (Supplementary Fig. 2b), show a pitting resistance similar to that of the polished LPBF material[21] and well above that of the polished conventional 316L SS[19,23]. While this highlights the excellent properties of the LPBF material as-built surfaces, it also means pits formed under high polarization currents. Therefore, we conducted measurements at OCP in 0.6 M NaCl on both top and side surfaces to be more representative of normal conditions (Supplementary Fig. 2c). Side surfaces experienced more metastable pitting than top surfaces at OCP, which agrees with the lower pitting potential than top surfaces (Supplementary Fig. 2b). After the OCP tests, pits were found to occur at slags (Fig. 3), regardless of the surface orientation; this confirms pit nucleation sites under these conditions follow the same pattern as they do under high polarization currents. Figure 3a, b shows an example of a pit that occurred at a slag. The SEM/EDS elemental maps (Fig. 3c) confirm the pit nucleation site is a Mn- and Cr-rich silicate slag. The more moderate conditions during OCP allowed us to capture pits at an early stage of formation on both top and side surfaces; all detectable pits have formed inside slags (Supplementary Figs. 3 and 4). In addition, more pits were visible on the side surfaces, which is consistent with the lower pitting potential and higher frequency of metastable pitting.

We point out that surface roughness does not play a significant role in determining the LPBF 316L SS pitting behavior in NaCl. Although the top surface, which has a lower roughness than the side surface (Supplementary Fig. 5), does offer higher resistance to pitting in our experiments, it has been shown previously that surface roughness does not correlate with pitting potential for LPBF 316 L SS[41]. In addition, we clearly showed that pits nucleate at slags on both surfaces during both OCP and potentiodynamic polarization testing, regardless of the surface roughness. The slightly lower pitting potential of the side surface therefore likely originates from the higher number of discontinuous and cracked slags rather than differences in roughness.

Based on our observations, we divide slags observed on LPBF 316L SS into two categories: Type I slags that are formed during the process and Type II slags that originate from the feedstock. Type II slags remain on the LPBF parts in regions that underwent only partial melting after exposure to high temperatures. The geometry and location of Type I slags differ on the top and side surfaces: on the top, they are elongated while on the side, they are smaller and randomly distributed. For convenience, these two subtypes will be referred to as Type I top and Type I side slags throughout the text. On the other hand, Type II slags appear spherical and are on the order of a few micrometers in diameter. Importantly, pits appear to initiate only at Type I slags.

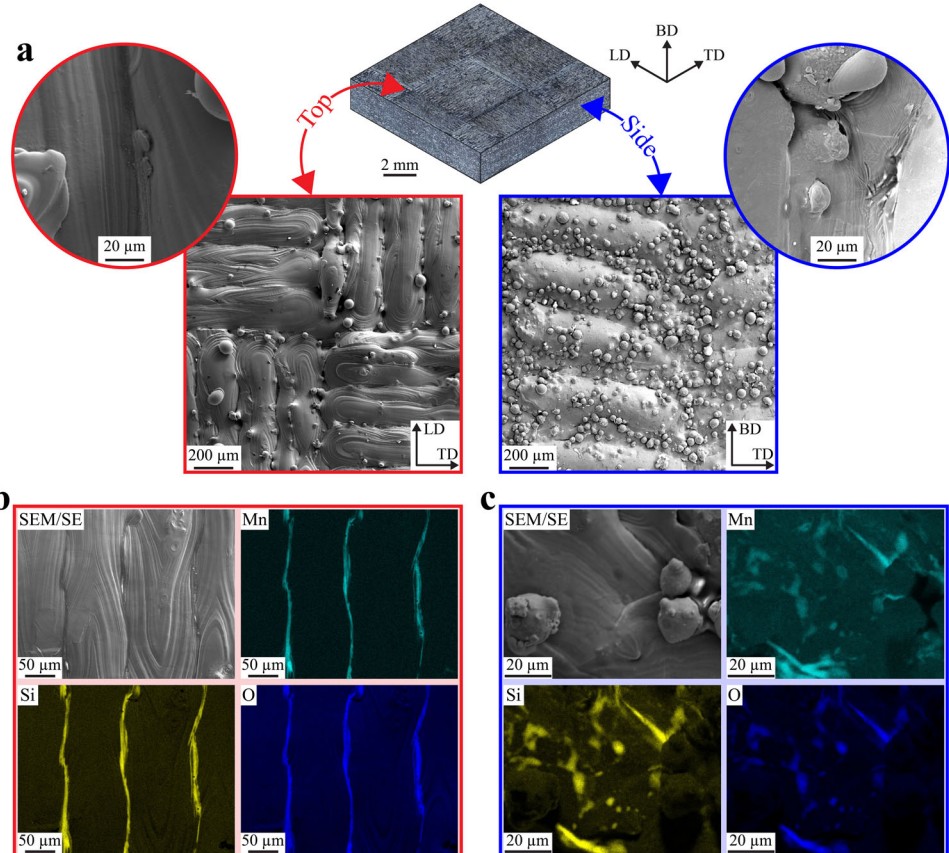

**Fig. 1 | Characterization of as-built surfaces of a LPBF 316L SS plate. a** Scanning electron microscopy (SEM) images, taken with secondary electron, showing the surfaces topography for two different surface orientations. The insets show higher magnification images highlighting defects. The SEM images are referred to an optical images montage to show their respective orientation compared to the build direction. **b**, **c** SEM energy dispersive X-ray spectroscopy elemental maps for the top and side surfaces, respectively. Elemental maps for Fe, Cr, and Ni are presented in Supplementary Fig. 1a and b, respectively.

A closer look at Type I slags using TEM is presented in Fig. 4. The location of the focused ion beam (FIB) lift-out is shown in Supplementary Fig. 6a. The scanning transmission electron microscopy (STEM) high angle annular dark field (HAADF) image in Fig. 4a shows a cross section of a Type I top slag. Selective area electron diffraction (SAED) images confirm that the slag is amorphous while the 316L SS is crystalline, and STEM/EDS maps in Fig. 4b further show that the slag comprises Cr-enriched Mn silicate. The full elemental composition of the slag is reported in Supplementary Table 1. Inside the slag, spherical inclusions can be observed. These are composed of Fe, Cr, and Ni as reported in Supplementary Table 2 and their size ranges from around 10 to 100 nm. Higher resolution STEM/EDS maps (Supplementary Fig. 7a, b) show the absence of Mn, Si, and O signal where these inclusions are located. They seem to have erupted from the metal/slag interface. Supplementary Fig. 7a, c clearly shows the initial stage of formation of one of these inclusions. This type of inclusion has been reported to form at the steel/slag interface or the slag/air (atmosphere) interface[42]. It has been postulated that rising gas bubbles through the slag drag steel droplets from the bulk steel into the slag[42].

Alone, these particles do not seem to be detrimental; however, when a large particle erupts at the edge of the slag, it can provide a direct pathway for the solution to access the unprotected metal (Supplementary Fig. 7a, d). For example, the results of a STEM/EDS line analyses L1 (shown on Fig. 4a) are presented in Fig. 4e. In the presence of a metallic inclusion near the edge of the slag of a thickness equivalent to the depth of the slag, the metallic inclusion and the 316L SS are only covered by a layer of $FeO_x$, and the protective Cr oxide layer usually found at the surface is absent. Locally, this composition offers little resistance to chloride solution attack, providing easy access to the bulk metal. Interestingly, the slag is also covered in a surface oxide (Supplementary Fig. 7a, b) composed mainly of Fe, Cr, Ni, Zn, and O. Note that the O signal overlaps well with Zn, but not with Fe, Cr, or Ni. In addition, the three latter elements appear to be dispersed in spherical inclusions at the surface rather than in a continuous layer. Accordingly, we conclude that the slag surface oxide is a Zn oxide with metallic inclusions. SEM images presented in Supplementary Fig. 8a, b show non-spherical inclusions just below the surface, likely corresponding to the metallic inclusions observed by TEM. It is important to note that the surface oxide is disrupted at the edge of the slag (Supplementary Fig. 7a, e) where it meets the metal surface oxide. At this interface between surface oxides, the oxide layer is locally much thinner (as clearly shown by the O signal in Supplementary Fig. 7e), which could also provide easier access to the bulk metal by the chloride solution.

In general, Type I slags present on the side surfaces show similar features as the ones located on the top surfaces. As an example, Fig. 4c shows TEM analyses of a cross-sectioned Type I side slag. Similar to Fig. 4a, the slag is amorphous, covered by an oxide layer, and contains crystalline spherical inclusions rich in Fe, Cr, and Ni (Fig. 4d and Supplementary Fig. 9a, b). However, there are some notable differences. The elemental composition of the metallic inclusions is slightly different from the ones found in the Type I top slags, as shown in Supplementary Table 2. Also, Type I side slags contain $CrO_x$ square shape inclusions; usually located at the bottom of the slag, they are sometimes observed below the surface (Fig. 4d and Supplementary Fig. 9a, b). Atomic resolution TEM (inset in Fig. 4c) reveals these inclusions to be crystalline with a large lattice parameter and ordered lattice substructures, likely a spinel structure[36]. This contrasts with the structure

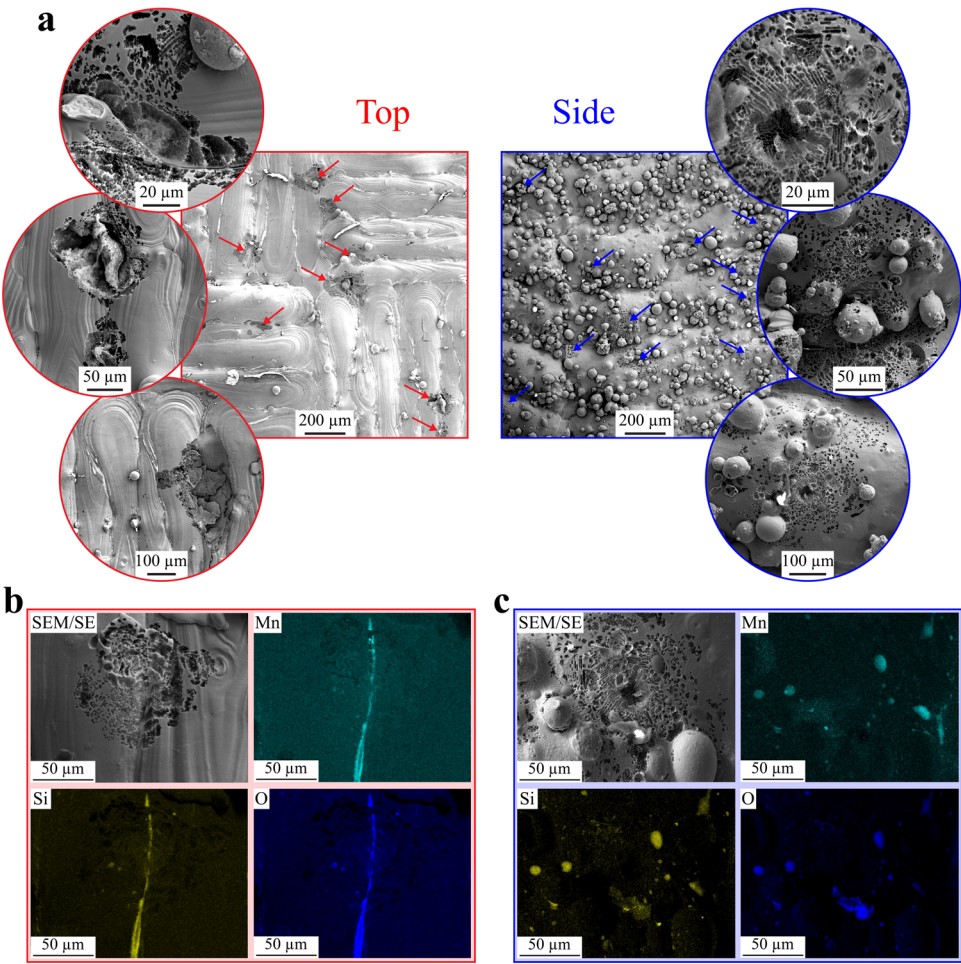

**Fig. 2 | Pitting corrosion of as-built surfaces. a** Scanning electron microscopy (SEM) images, taken with secondary electron, of the as-built surfaces after polarization testing. Insets highlight various pits at the surfaces. Red and blue arrows point at pits. **b, c** SEM energy dispersive X-ray spectroscopy elemental maps for the top and side surfaces, respectively. Elemental maps for Fe, Cr, and Ni are presented in Supplementary Fig. 1c and d, respectively.

of the spherical inclusions, which appear closer to fcc austenite (see corresponding atomic resolution TEM image in the inset in Fig. 4c). The presence of Cr in slags has been previously observed in conventional stainless-steel literature, as Cr oxide is thermodynamically preferred over Fe and Ni oxides at steelmaking temperatures[42]. Therefore, chromium in molten steel can be preferentially oxidized and end up in the slag phase. Another difference between Type I top and side slags is the composition of the surface oxide. There still is a continuous Zn oxide, but it seems to be overlapping with an Fe oxide layer and there is no Cr (Fig. 4d and Supplementary Fig. 9a, c). In some places, small Si inclusions can also be seen.

Type I slags, both top and side, exhibit cracks, as shown by SEM in Supplementary Fig. 10. Observed in Fig. 4c, d and studied in more details in Supplementary Fig. 9d, e, cracks form in the slags due to thermomechanical stresses during cooling. Cracking seems to appear at a stage when most surface oxides are already formed. As a result, at the bottom of the crack, the metal is exposed to the corrosive solution with almost no protection (Supplementary Fig. 9d). STEM/EDS line analyses (Fig. 4f) reveal that the only surface oxide present at the bottom of the crack, on top of the metal, is a layer of Fe oxide, and no protective Cr oxide. This would explain why pits seem to form inside slags in Figs. 2 and 3 and Supplementary Figs. 1, 3, and 4.

From discontinuities at the edges to cracks all the way through, it is clear that Type I slags contain several defects that can ultimately allow corrosive solutions to access the metal and initiate pitting. Looking at LPBF 316L SS surface cross-sections after corrosion can shed additional light on the role of these slag defects in pitting corrosion. In Fig. 5, we show a cross section of a Type I slag after potentiodynamic polarization testing in 0.6 M NaCl. Figure 5a to e present successive SEM images through a pit at the Type I slag. We can clearly see that the slag remained while the metal underneath dissolved during pit propagation. Figure 5a shows the slag near the pit. Figure 5b highlights (red circle) a region where the slag is discontinuous, and a point of access for the NaCl solution is visible. Figure 5c, taken further into the pit, highlights the differences in dissolution rate due to the rapid solidification cellular structures, as we previously observed[30]. Here, cell walls dissolved faster than cell interiors, which has been attributed to the activation of transpassive dissolution of Cr under high potential during polarization testing[43]. Figure 5d depicts the edge of the slag that delaminated, suggesting the metal/slag interface can be weakened by the corrosive solution. Figure 5e focuses on a metallic inclusion inside the slag that dissolved, demonstrating that these inclusions are a potential pathway for the solution to travel though the slag when inclusions are found near the interface. Note that cracks were not present in the cross-sectioned region. We used TEM to look at the bottom of the crack in Fig. 5f. The shape of the metal surface at the bottom has changed due to dissolution in the chloride solution. The metal/slag interface likewise appears affected, especially on the left-hand side. STEM/EDS maps (Fig. 5g) show no Cr at the exposed metal at the bottom of the crack, highlighting the absence of passivation. This confirms that cracks are a weak point for corrosion as the metal is left without protection.

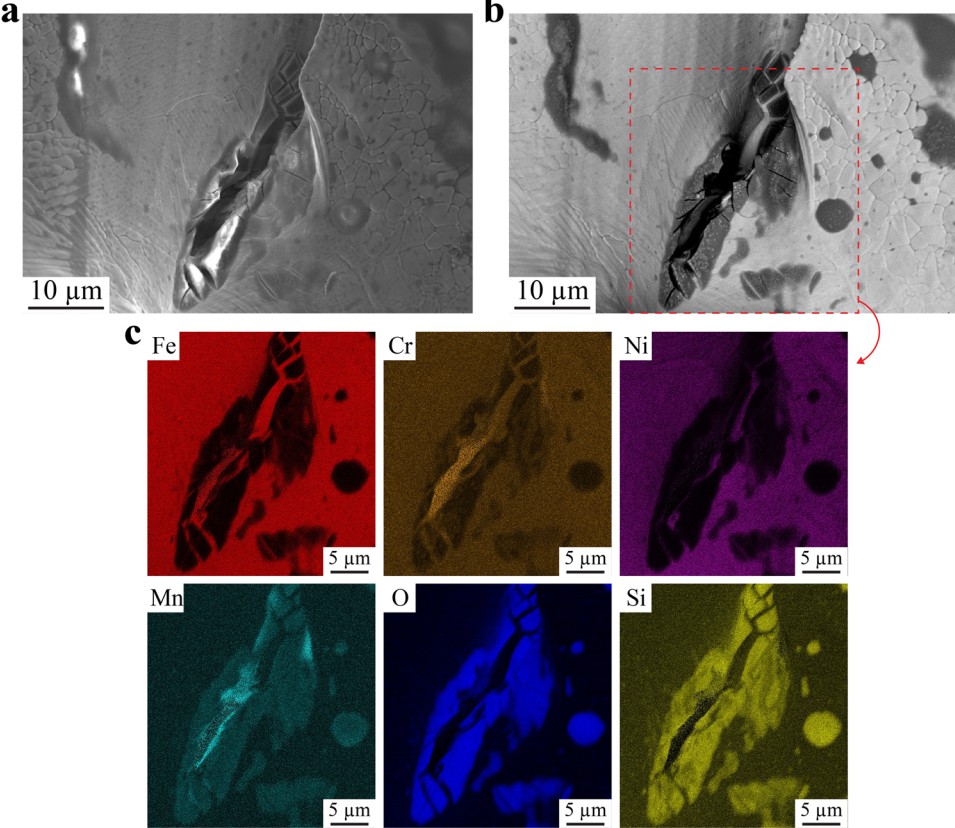

**Fig. 3 | Pit at a slag after open circuit potential testing in 0.6 M NaCl.**
**a**, **b** Scanning electron microscopy (SEM) images of a same pitted slag obtained with secondary electrons and back-scattered electrons, respectively. **c** SEM energy dispersive X-ray spectroscopy of the regions highlighted with the red, dashed square in (**b**).

Differences in metal composition are a potential concern for galvanic corrosion. Along these lines, a significant increase in Cr at the interface between 316L SS and Type I slags can be observed (Supplementary Fig. 11). The liquid metal can accept more Cr than the solid 316L SS; as such, upon rapid solidification during LPBF, Cr oversaturates the solid that cannot quickly homogenize, and the surplus Cr is pushed into the liquid until it reaches the last place of solidification. Consequently, similarly to Cr trapping in the cellular structures[44], there is a higher concentration in Cr near the slag. STEM/EDS line scans (Supplementary Fig. 11a, b) show the fraction of Cr can increase up to $\approx$ 30 at. % at the metal/slag interface, with an associated decrease in Fe and Ni. This is much more than the increase usually observed in the cellular structures[45]. This could cause the creation of a local galvanic cell between the metal and the Cr-rich region[46], which could further accelerate corrosion at the slag interface. Note that this same hypothesis cannot apply to Type II slags, which do not exhibit a Cr gradient in the metal at the slag/metal interface (Supplementary Fig. 11c) because they do not experience melting during LPBF.

In contrast to Type I slags, Type II slags do not pit, which can be attributed to the absence of defects and the presence of a protective layer of alumina. Presented in Fig. 6, TEM characterization of a FIB cross section lifted out from the region highlighted in Supplementary Fig. 6c shows that Type II slags have a different shape, in the form of an equiaxed convex lens. More importantly, unlike Type I slags, there is a quasi-absence of inclusions and cracks. In addition, a thick aluminum oxide layer entirely covers the slag (Fig. 6a) and homogeneously continues onto the rest of the neighboring metal (Fig. 6b). This aluminum oxide layer is mixed with some Fe and Si oxides, which provide extra protection, and is not found elsewhere in the material. Together, the presence of Al oxide and lack of defects results in Type II slags

appearing unaffected after polarization testing (Fig. 6c, d). Although the surface near Type II slags seems to have sustained intergranular attack (Fig. 6c), this pattern did not develop under corrosion but during LPBF; it is observed on the as-built, untested surfaces (Supplementary Fig. 12). The underlying observed microstructure is coarse (Fig. 6d), without the characteristics of the rapidly solidified metals. This indicates that Type II slags belong to un-melted regions, and as such, formed on the feedstock.

To better understand why Type I slags are the preferred pit nucleation sites upon exposure to chloride solution, it is critical to contrast their behavior with that of the intact native metal surface oxide covering most of the as-built LPBF parts apart from the slags. TEM analyses of a cross-sectioned as-built laser track, away from slags, are presented in Fig. 7. The surface oxide appears to be thick, ranging from 20 to 200 nm (Fig. 7a). Interestingly, no correlation between the oxide thickness or chemical composition could be made with the underlying microstructures. The surface oxide is composed of three layers (Fig. 7b, c). From the metal to the surface, the first layer is $MnSi_xO_x$, the second is a mix of $CrO_x$ and $FeO_x$, and the third is a Zn oxide. These layers are not abruptly separated but rather are interconnected. XPS measurements (Supplementary Fig. 13) confirm the presence of Fe, Cr, Zn, and Si oxides on the material surfaces. The inner layer of $MnSi_xO_x$ suggests this oxide was formed at high temperature, as it is typically seen in high-temperature oxidation of Fe–Cr steels with Si addition[47].

Notably, the inner layer of $MnSi_xO_x$ is not affected by the chloride solution. To see this, the native surface oxide was investigated after corrosion testing (OCP and polarization) in 0.6 M NaCl. Figure 8 shows the surface oxide after 35 days of continuous testing at OCP (corresponding OCP curves are in Supplementary Fig. 2c). STEM/EDS

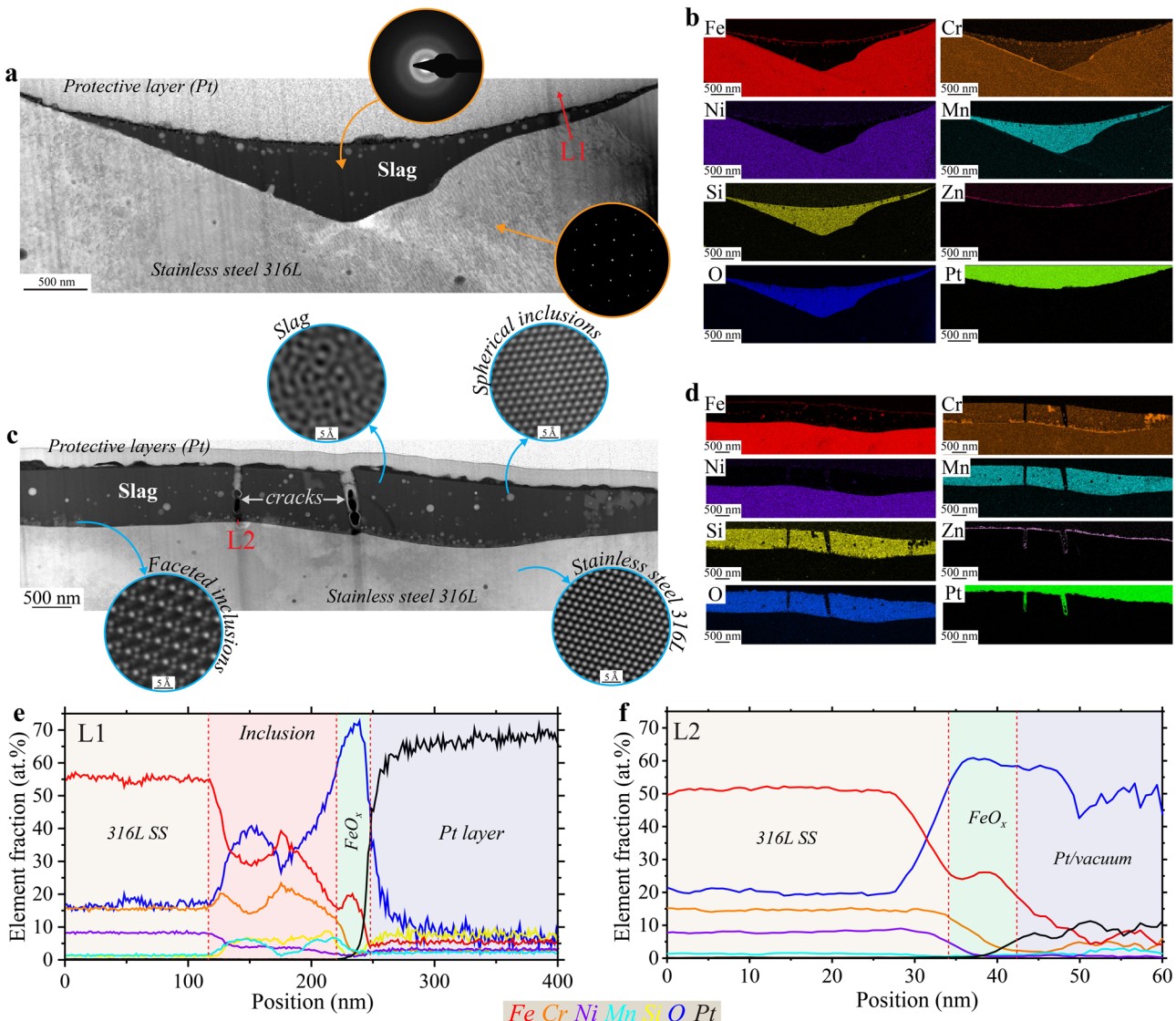

**Fig. 4 | Transmission electron microscopy characterization of Type I slags found on top and side surfaces. a, b** Scanning TEM (STEM) high angle annular dark field (HAADF) and STEM energy dispersive X-ray spectroscopy (EDS) maps, respectively, conducted in the same region of a cross-sectioned Type I top slag. The region where the TEM sample was lifted out with focus ion beam is presented in Supplementary Fig. 6a. **c, d** STEM/HAADF and STEM/EDS maps, respectively, conducted in the same region of a cross-sectioned Type I side slag. The region where the TEM sample was lifted out by FIB is presented in Supplementary Fig. 6b. **e, f** STEM/EDS line analyses corresponding to L1 (**a**) and L2 (**c**), respectively. The color legend applies to both (**e, f**).

elemental maps (Fig. 8a) reveal a surface oxide with a composition that appears to be largely maintained after testing, composed of four different oxides: $MnSi_xO_x$, $CrO_x$, $FeO_x$, and $ZnO_x$. However, the STEM/EDS line analysis through the surface oxide (Fig. 8b) highlights some differences. In particular, some Mn and Zn have diffused through the different oxide layers, and the Cr oxide has partially dissolved. Despite these changes, the $MnSi_xO_x$ remains unchanged, covering and protecting the metal surface. A similar picture emerges upon more aggressive potentiodynamic polarization testing. Figure 9a shows a STEM/HAADF image and corresponding elemental maps of the native oxide after polarization testing (corresponding polarization curves are in Supplementary Fig. 2a). Figure 9b is a STEM/EDS line analysis through the native oxide. Most of the Cr oxide layer appears to have been dissolved by the chloride solution, but the layer of Fe oxide covered by another layer of Zn oxide remains on top of the degraded Cr oxide. This is due to the transpassive dissolution of Cr under high polarization potential[43] as also observed inside pits at the cell walls (Fig. 5c). Nevertheless, the first layer of Mn silicate again remains

intact, still protecting the metal below. It is crucial to note that although the chemical compositions of slags can resemble this silicate layer, the layer remains continuous and not cracked. This is in part because it is very thin and forms directly at the solid metal surface, whereas slags form inside melt pools and solidify in regions exposed to residual stresses. In conventional steels, it has been shown that this silicate phase is protective in corrosive solution and slows down chloride diffusion occurring through the Fe/Cr oxide layer above[47]. We conclude that the high resistance to pitting in NaCl of the native surface oxide of as-built 316L SS surfaces likely comes from the presence of this continuous Mn silicate layer.

Having established that slags are the preferential nucleation sites for pits, it is useful to understand how they can reach the surface during LPBF. Unlike casting or welding, melt pools in LPBF are very dynamic. For example, in stainless steel welding, rising of the slag from the inside of the melt pool to the outside surface is mainly driven by the Marangoni flows due to differences in surface tension and temperature gradients[31–33]. In LPBF, melt pools are significantly smaller, as

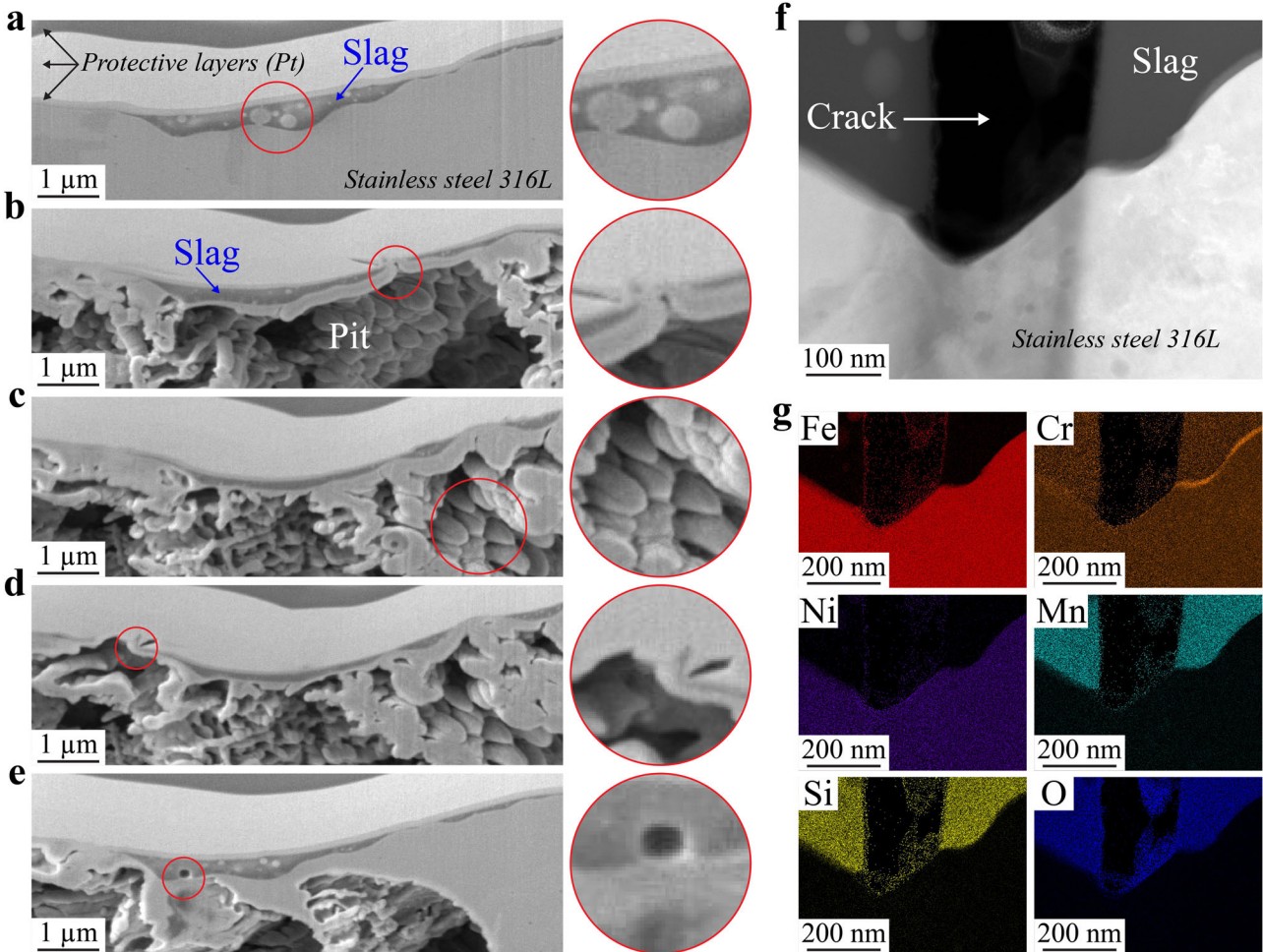

**Fig. 5 | Characterization of Type I slags after corrosion testing. a–e** Slice and view scanning electron microscopy (SEM) images after successive plasma focus ion beam milling of a pitted Type I slag. All insets are higher magnification images of the regions marked with red circles. Each inset highlights specific feature: **b** A discontinuity of the slag where the NaCl solution was able to penetrate the metal, **c** Solidification cells after metal dissolution, **d** The edge of the lag that delaminated. **e** A metal inclusion in the slag that dissolved due to the solution. **f** Scanning transmission electron microscopy (STEM) high angle annular dark field and **g** STEM energy dispersive X-ray spectroscopy maps of a crack tip in a corroded Type I slag that did not pit.

such, recoil pressure also plays a significant role in controlling flows in the melt[48,49]. This creates a more violent melt pool, which strongly affects flow direction. Understanding slag formation therefore requires understanding liquid flows in LPBF melt pools. In Fig. 10, we use a three-dimensional high-fidelity model capable of including recoil pressure to simulate the thermal profile and liquid flows in a melt pool. Using a ray-tracing technique, this model accounts for powder particles contact regions and enables partial particle melting for a more accurate volumetric energy deposition[48,49]. Figure 10a shows the initial powder-bed before the laser is turned on. In Fig. 10b, the laser was just turned on and the melt pool starts forming. The arrows point toward the direction of the local liquid flow, and their colors indicate relative flow velocities. At this stage, the liquid flow is directed to the melt pool surface, dragging any formed slag to the top. Figure 10c is taken at a later stage of the melting when the melt pool is the deepest. It can be clearly seen that the liquid is still flowing to the surface. In addition, the flow changes direction when reaching the surface, turning to the sides where Type I top slags were found. In Fig. 10d, the laser has progressed, and the melt pool rapidly solidifies starting from the edges, which traps slags. In Fig. 10e, f, the solidification process progresses to completion. This liquid flow simulation supports well the notion that slags should be found at the surface along laser track edges. Not only does the lower density of the slags naturally force them towards the surface, but the melt pool internal flows also favor their trapping at the

melt pool edges. Slags are found mainly at the as-built surfaces, as confirmed in the EDS elemental maps in Supplementary Fig. 14a, b (only in a few instances did we find slags inside the bulk, away from the surfaces; Supplementary Fig. 14c, d). Although slags can form at every layer during LPBF, their near absence inside the bulk suggests that the remelting occurring with each layer is sufficient to dissolve the slags until the surface is reached, at which point no more remelting occurs.

In this work, we found Mn-rich silicate slags to be responsible for LPBF 316L SS pitting corrosion in chloride solution. Supplementary Fig. 15 summarizes the different slags and native protective oxides that were found at the as-built surfaces. Their dimensions, structure, location, and level of protection are all related to the temperature and environment in which Mn and Si reacted with oxygen. Note that Mn and Si are usually introduced in the 316L SS composition to limit the amount of oxygen in the metal during conventional casting or welding. However, LPBF is a different process due to the nature of the feedstock and extreme local processing conditions. The merits of Mn and Si for LPBF 316L SS are therefore less clear and prompt further investigation. In particular, we propose that a new 316L SS composition could be adapted to LPBF by avoiding the addition of Mn and Si to prevent slag formations. As alloy design specifically for additive manufacturing emerges as a new field[50], it is obvious that corrosion resistance should factor as an optimization parameter. We further understand now that the surface roughness is not a proper metric to assess corrosion when

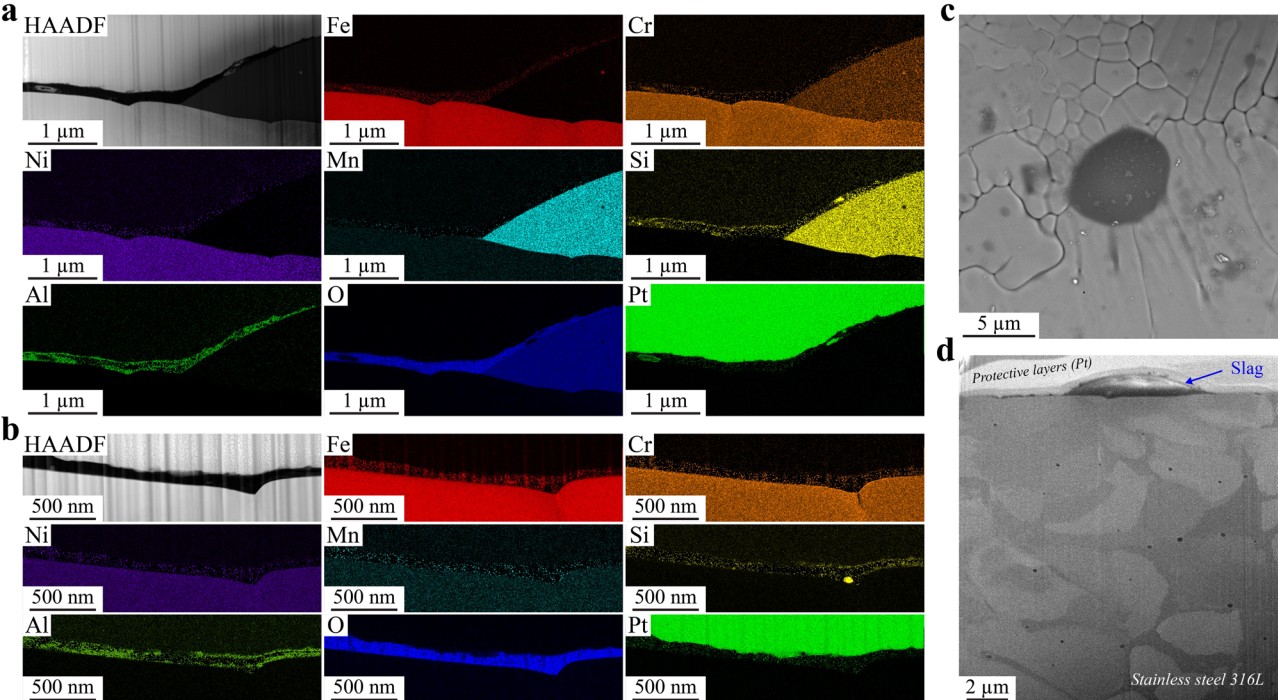

**Fig. 6 | Transmission electron microscopy (TEM) characterization of a Type II slag found on an un-melted surface.** Scanning TEM (STEM) high angle annular dark field (HAADF) and corresponding STEM energy dispersive X-ray spectroscopy elemental maps of the slag and of the native surface oxide away from the slag are shown in (**a**) and (**b**), respectively. The region where the TEM sample was lifted out by focus ion beam is presented in Supplementary Fig. 6c. **c**, **d** A top view and a cross section view of a Type II slag after corrosion testing.

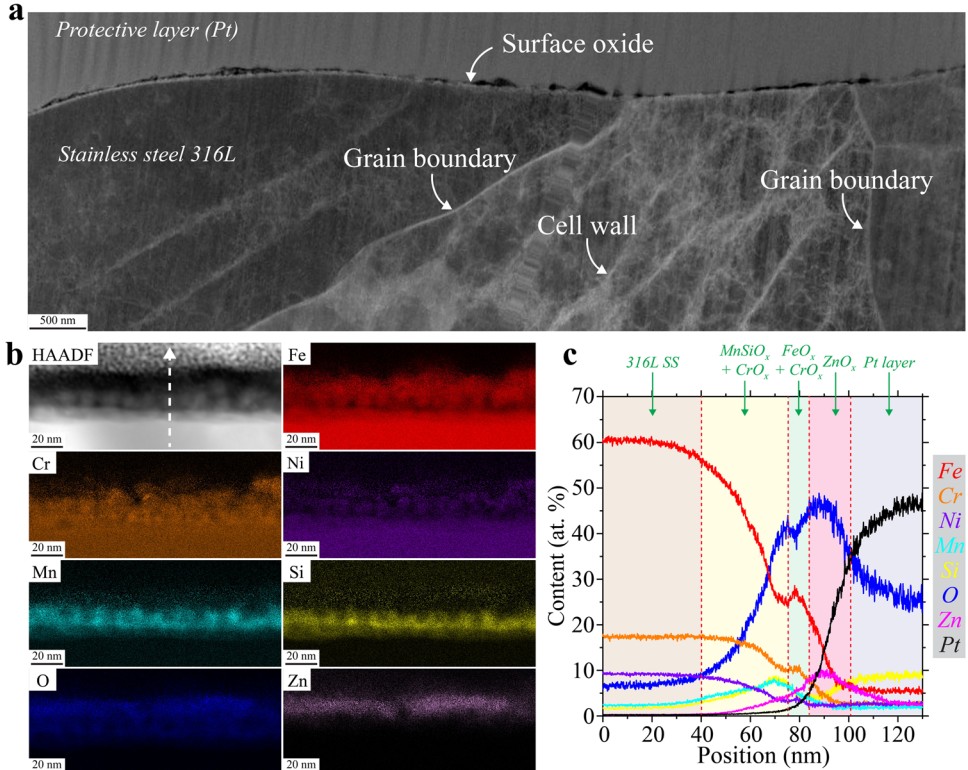

**Fig. 7 | Transmission electron microscopy (TEM) characterization of the native surface oxide found away from slags or partially melted regions. a**, **b** Scanning TEM (STEM) high angle annular dark field (HAADF) and STEM energy dispersive X-ray spectroscopy (EDS) maps, respectively, conducted at the surface of a cross-sectioned laser track of the top surface. **c** STEM/EDS line analyses conducted along the white dashed arrow noted on (**b**).

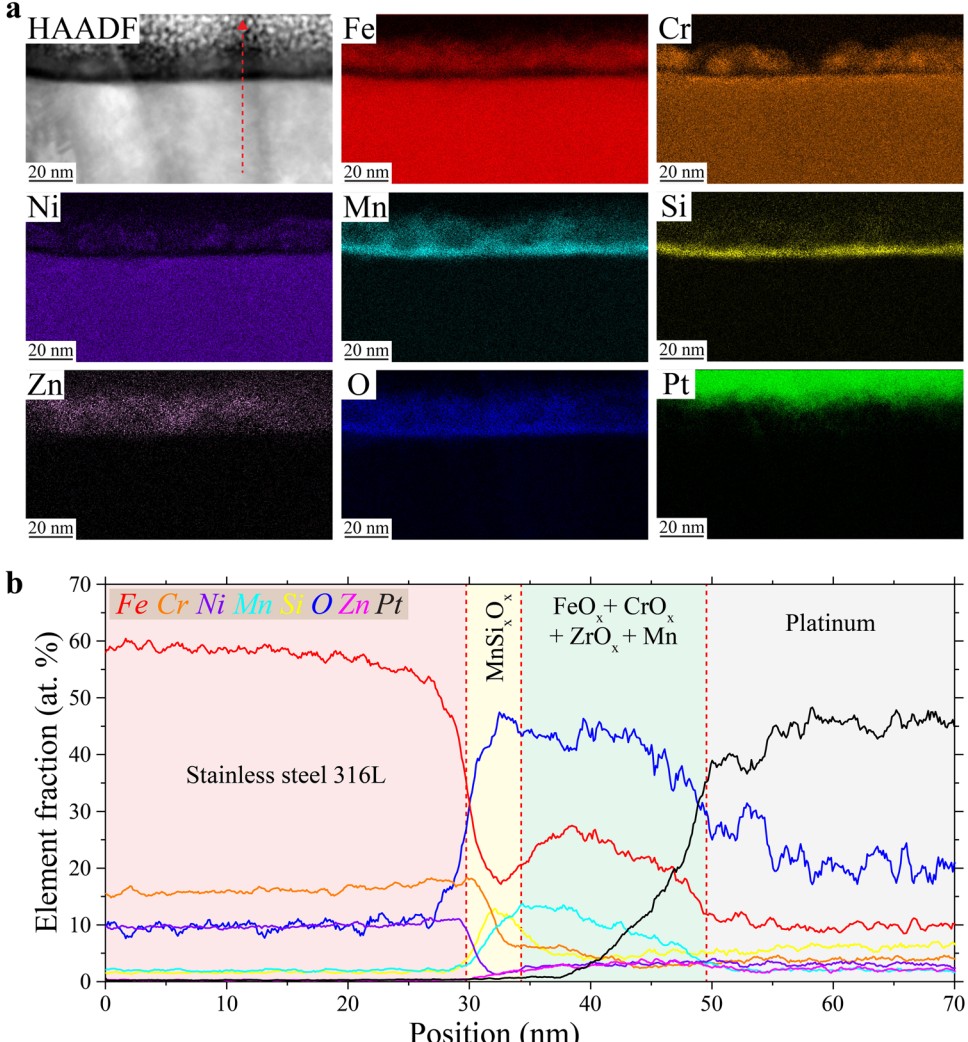

**Fig. 8 | Transmission electron microscopy (TEM) characterization after open circuit potential testing of the native surface oxide found away from slags or partially melted regions. a** Scanning TEM (STEM) high angle annular dark field (HAADF) and STEM energy dispersive X-ray spectroscopy (EDS) maps of the corroded surface oxide. **b** STEM/EDS line analysis along the red dashed line drawn in (**a**).

rapid cooling rates in LPBF modify secondary phases responsible for pitting. Instead, LPBF alloys compositions need to be re-evaluated with new perspectives in terms of both corrosion and mechanical properties. For example, our observations of the apparent role of aluminum in partially melted regions, where protective layer formation prevented pitting at the slags, opens additional possibilities for adapting alloy compositions or developing artificial coatings to protect entire parts.

In summary, this work identified the origin of pitting corrosion on as-built surfaces of LPBF 316L SS and established a link with processing conditions and local microstructures. We showed that unique conditions developed during LPBF give rise to new mechanisms and potentially better properties. At the same time, our work indicates that there is significant room for improvement. Particularly compelling is the possibility of co-designing the feedstock alongside the laser processing parameters to finely tune local microstructures with improved chemical and mechanical robustness.

## Methods
### Material manufacturing
All 316L stainless steel material used in this work was manufactured by laser powder bed fusion using a Concept M2 machine with a laser power of 150 W and scanning speed of 700 mm/s[8,45]. Layer thickness

was 30 μm and hatch spacing was 105 μm. We used an island scanning strategy with 5 × 5 mm² islands. The scan vector was rotated 90° between the neighboring blocks. This pattern was rotated by 90° every layer and shifted along x- and y directions by 200 μm. All material was studied as-built; i. e., no post processing such as machining, polishing, or heat treatment was performed before testing and characterization.

### Surface roughness measurements
Surface roughness measurements of as-built parts were carried out using a Zygo 7300 White Light Interferometer.

### Electrochemical characterization
Electrochemical characterization was carried out using Biologic VSP300 potentiostat. A three-electrode configuration was used with the samples used as working electrode, graphite used as the counter electrode, and a saturated calomel electrode (SCE) served as reference electrodes. To reduce the electrical measurement noise during testing, all electrochemical cells were positioned inside a grounded Faraday cage. Before and after electrochemical testing, samples were cleaned and rinsed with distilled water, and subjected to an ultrasonic bath containing analytical-grade isopropyl alcohol for 5 min before being dried with flowing nitrogen gas. Samples were submerged in ~300 ml 0.6 M (3.5 wt.%) NaCl testing solution (representative of seawater) for

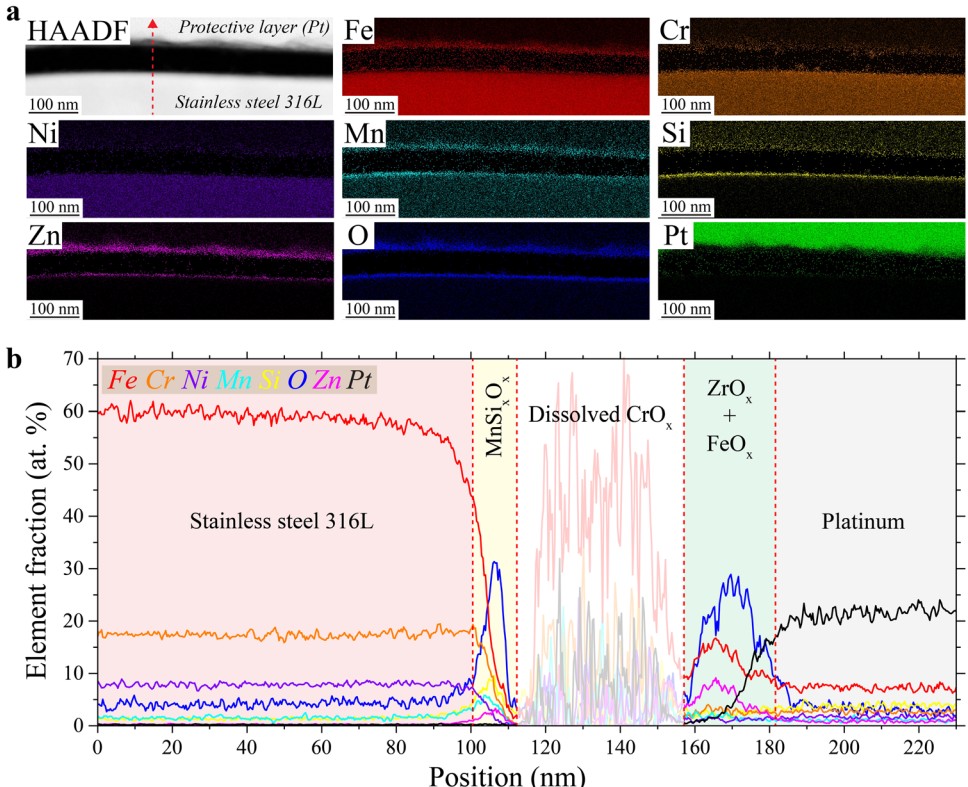

**Fig. 9 | Transmission electron microscopy (TEM) characterization after polarization testing of the native surface oxide found away from slags or partially melted regions. a** Scanning TEM (STEM) high angle annular dark field (HAADF) and STEM energy dispersive X-ray spectroscopy (EDS) maps of the corroded surface oxide. **b** STEM/EDS line analysis along the red dashed line drawn in (**a**).

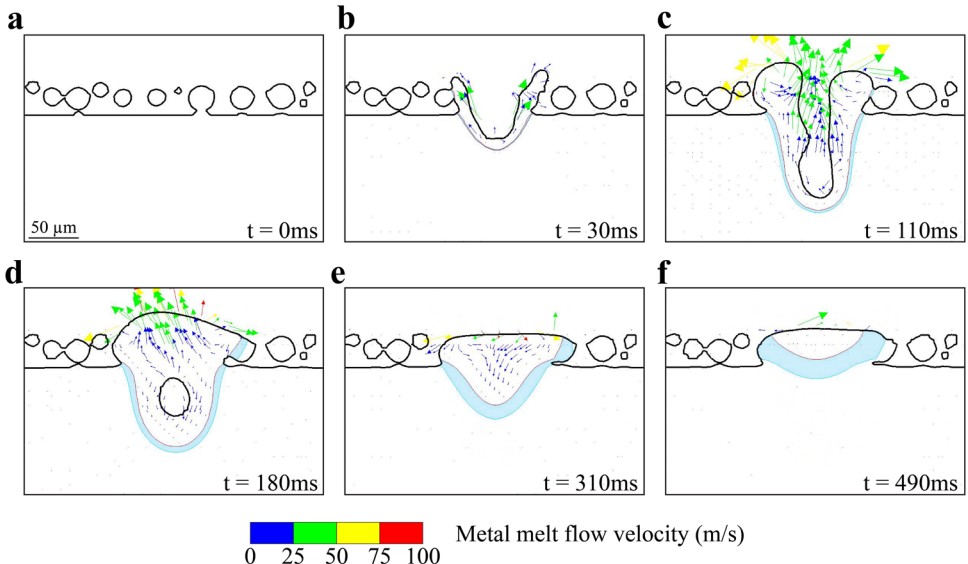

**Fig. 10 | Three-dimensional high-fidelity simulation of liquid metal flows in a stainless steel 316L melt pool caused by a laser raster on a powder bed. a–f** The evolution of the melt pool as the laser advances on the surface. The blue region highlights the area where the local temperature is below the solidus of stainless steel but potentially above the liquidus of the slags. Arrows indicate the direction of the local liquid flow and their length scales with the flow velocity.

1 h before starting the potentiodynamic polarization test. Potentiodynamic polarization scans commenced from −100 mV below the open circuit potential to reach 10 mA/cm² or 1500 mV vs SCE whichever attains first using a scan rate of 10 mV/min. Each sample surface orientation (side and top) was tested ten times to ensure reproducibility and statistically meaningful results. Additionally, side and top surfaces were tested at the OCP in 0.6 M (3.5 wt. %) NaCl to gain insight into corrosion properties under natural, non-polarized conditions. The OCP was recorded for a period of 35 days.

**Focused ion beam sectioning**
Focused ion beam (FIB) cross sectioning and scanning electron microscopy (SEM) of as-built top and side surface samples was performed using a FEI NanoLab600i DualBeam™ FIB/SEM station.

Samples were prepared for sectioning by first mounting the samples on a SEM stub utilizing a pleco tab. The stub was then mounted into the FIB/SEM instrument and pumped down for high vacuum. A gas injector was inserted to perform an e-Pt (electron) chemical vapor deposition (CVD) protection layer followed by I-Pt (ion beam). After the deposition, a Ga+ ion beam was used to mill out the section and transferred the section to a TEM grid using a micromanipulator. After transfer, the section was milled to electron transparency using the Ga+ ion beam.

### P-FIB slice and view
Milling for slice and view analysis utilized a Helios 5 plasma FIB Dual-Beam with an inductively coupled plasma Xe+ beam operated at an accelerating voltage of 30 kV. A gas injector was inserted to perform an e-Pt (electron) chemical vapor deposition (CVD) protection layer followed by I-Pt (ion beam) prior to the start of milling.

### Electron microscopy
Transmission electron microscopy (TEM) using bright-field (BF), high angle annular dark field (HAADF) scanning TEM imaging (STEM), and STEM energy dispersive X-ray spectroscopy (STEM/EDS) was performed on a FEI Titan operated at 300 kV. STEM/EDS composition analysis was performed with the ThermoFisher SuperX ChemiSTEM system using the Velox software.

Scanning electron microscopy (SEM) imaging was performed with a ThermoFisher Apreo 2. SEM/EDS was conducted with a EDAX Elite Super EDS detector using the APEX™ software from EDAX.

### XPS characterization
X-ray photoelectron spectroscopic analysis was performed using a Physical Electronics Quantum 2000 spectrometer equipped with a monochromated Al Kα source ($hv = 1486.6$ eV). As-built samples were placed on double sided carbon tape and mounted on silicon wafers for analysis. Calibration of the instrument was performed using Au $4f_{7/2}$ at 84.1 eV and the take-off angle was 60°. A 200 μm x-ray spot size was used. A base pressure of $10^{-9}$ mbar was maintained in the analytical chamber. Survey spectra for each sample were recorded using a pass energy of 100 eV and a step size of 1 eV, and high-resolution spectra of each sample were recorded with a pass energy of 20 eV and a step size of 0.1 eV. The C1s peak at 284.8 eV, for adventitious carbon, was used as a reference for all spectra. Two sets of survey and two sets of high-resolution spectra were measured for as-built top and side surfaces.

### Laser track simulations
We used a high-fidelity model that was previously validated against X-ray experiments[51,52]. The model relies on ALE3D[53] multi-physics code, developed at Lawrence Livermore National Laboratory and uses an Eulerian mesh. The model couples the hydrodynamics with the solution of the thermal diffusion equation and accounts for phase transformations using operator splitting. The heat source is provided by full laser ray tracing for accurate thermal budgeting. No calibration of the laser absorptivity is assumed. The model computes the material's absorptivity by accounting for the electric conductivity and the gas/material geometric interface. Further experimental validation of the absorptivity was performed on bare plate surface for SS316-L[54].

## Data availability
Data generated during the current study is available from the corresponding author on request.

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

## Acknowledgements

Authors were supported by the Laboratory Directed Research and Development (LDRD) program (20-SI-04) at Lawrence Livermore National Laboratory. This work was performed under the auspices of the US Department of Energy by Lawrence Livermore National Laboratory under contract No. DE-AC52-07NA27344.

## Author contributions

S.S-B. performed XPS measurements. T.V., S.S-B., and T.T.L. performed TEM characterizations. S.S-B., D.R, and T.V. performed SEM characterizations. V.S. performed P-FIB lift-outs and P-FIB slice and view characterizations. S. C. and Z.Q. conducted surface roughness and electrochemical measurements. S.K. developed laser tracks simulations. S.S-B., S.R.Q., B.C.W., and T.V. wrote the manuscript. T.V. and S. R. Q. led and guided the research. T.V., S.S-B., B.C.W., S.R.Q., S.C., Z.Q., T.T.L., D.R., V.S., S.K., Y.M.W., and S.K. contributed to the data analysis, discussion, and review of the final manuscript.

## Competing interests

We declare that none of the authors have competing financial or non-financial interests as defined by Nature Portfolio.
