## [Peer Review File · Nature Communications]

Critical role of slags in pitting corrosion of laser powder bed fusion additively manufactured 316L stainless steel in simulated seawaterREVIEWER COMMENTS

Reviewer #1 (Remarks to the Author):

Nature Comm Review report: Critical role of slags in pitting corrosion of laser powder bed fusion additively manufactured 316L stainless steel in salt water

This manuscript reports a study on the origin of pitting corrosion on as-built surfaces of LPBF 316LSS. The main finding is "that Mn-rich silicate slags are responsible for pitting of the as-built LPBF material in sodium chloride due to their introduction of deleterious defects such as cracks or surface oxide heterogeneities". It is claimed "that LPBF changes surface oxides due to rapid solidification and high temperature oxidation, leading to fundamentally different pitting corrosion mechanisms".

Although this is an interesting and welcome study on the localised corrosion behaviour of LPBF 316LSS, the reviewer thinks that care needs to be taken in several aspects of the work:

1. The identification of pitting corrosion of as-built SS316L surfaces was done on as-built surfaces after polarization testing in NaCl solution. It is not surprising to see anodic dissolution occurred on defects such as cracks or surface oxide heterogeneities in slags because these defects provide the lowest resistant path for polarisation currents. This is the same as polarising a metal surface with porous organic coatings – polarisation currents would certainly pass through pores in the organic coating because of much lower resistivity. In practical/natural corrosion condition under open circuit potential (OCP), however, the actual pitting initiation may be different. Although these defects in slags would still provide easy paths for electrolyte transportation, they may not be the most critical factor affecting pitting, and the actual pitting processes and mechanisms may be different. The reviewer thinks that there is need for corrosion tests at OCP in NaCl and/or FeCl₃ solutions. These tests may take a lot longer to do, but they will reveal pitting processes and mechanisms under practical corrosion conditions.

2. On the other hand, since the pitting testing was on the as-built surfaces after polarization testing, not under OCP, care needs to be taken when claiming the finding of 'fundamentally different pitting corrosion mechanisms'.

The reviewer strongly suggests that the authors should perform pitting corrosion tests at OCP in order to ensure their key findings apply to natural pitting processes.

Nevertheless, I reviewer thinks that this is an interesting paper and worth considering publication in Nature Communications

Mike Yongjun Tan, PhD, FNACE
Deakin University, Australia

Reviewer #2 (Remarks to the Author):

This manuscript provides an extensive characterization of two different types of slag that become trapped in LPBF 316LSS. The TEM work of the authors is exemplary, and their analyses of the chemical compositions of the slags and surface films on the 316L provide important data for understanding the as-printed surface condition of AM components. The CFD work is outside my area of expertise, so I will focus on the interpretation with regards to pitting corrosion. Below I have followed the questions provided in the request for review

- What are the noteworthy results?

1. Identification and characterization of the two types of slag along with strong evidence supporting different origins.
2. This may reveal my ignorance of AM processing of SS316L, but I was surprised by the presence of zinc and aluminum oxides on the as-printed surfaces at and, in some cases, near the slag. I could not find what the authors believe to be the origin of these impurities, but their presence is undeniable.
3. The correlation between Mn- and Si-bearing slags and corrosion, if substantiated, would be an important guide to powder composition development, as pointed out by the authors.

- Will the work be of significance to the field and related fields? How does it compare to the established literature? If the work is not original, please provide relevant references.

The TEM work, especially the compositional analyses, of the slags is important. I strongly agree with the authors that as-printed surfaces are generally ignored in studies of AM corrosion, although the design versatility of AM will certainly lead to such surfaces that are inaccessible for post-build surface finishing (e.g., inside tubular structures). In addition, the observation of the formation of what seems to be a tenacious aluminum oxide on the surface could guide alloy development by purposely including it in the alloy powder.

- Does the work support the conclusions and claims, or is additional evidence needed?

The work is consistent with the claims, but does not prove them. To do so would require capturing the pitting at a much earlier stage than shown in Figure 2 in which the pits are far too numerous and large to attribute initiation to the presence of the Type I slag.

- Are there any flaws in the data analysis, interpretation and conclusions? - Do these prohibit publication or require revision?

The argument that cracks in the slag oxide are what lead to preferential pitting is questionable. While Figures 3, 5, S4, and S6 do show cracks in the oxide that reach the SS surface, but the images in Figure 2 and 4 do not convince me that the pits start at the slags. The level of damage makes it impossible to prove the initiation point. The fact that the pits grow UNDER the slag is not necessarily odd; the slag acts like a crevice former. In the case of Type II slags, the alumina layer formed may well protect that area of the surface.

Maybe I missed it, but as the slag are created during deposition, should there be stringers of them throughout the part as each layer would have its own slag? Or is remelting that occurs upon deposition of subsequent layers sufficient to completely dissolve the slags? Wouldn't some of these slags be surrounded by molten metal, but not be melted? If so, polished surfaces would ALSO show pitting at the slags, in the authors' hypothesis is correct.

The authors do not mention what look like intergranular attack in Figure 5c. That would also be an important observation for a material that was tested as-printed without any sensitization heat treatment.

- Is the methodology sound? Does the work meet the expected standards in your field?

The approach used to create the corrosion is not optimal. Polarization scans in 0.1M NaCl may be a reasonable first step to characterizing corrosion behavior, but they are grossly insufficient as a means to represent the corrosion conditions to which any stainless steel component will be exposed. The concentration of chloride is too low, and this leads to the need for very high levels of polarization to initiate and grow pits ($> +0.5V(SCE)$ in all but one case shown in Figure S2). There is no natural or service environment in which such polarization can occur. The problem this level of polarization creates is in the morphology of the damage. At high polarizations, the high Cr in the cell walls is preferentially dissolved via transpassive dissolution, leading to the damage morphology shown in Figures 4c, for example. At lower potentials far more relevant to service environments (e.g., $< 0V SCE$), the interiors of the cells are preferentially attacked. The use of high polarization has been the bane of the corrosion field at times, and propagating it into the AM field would be unwise. This transpassive dissolution also applies to Figures 6 and 7 in which the disappearance of the Cr oxide, but retention of iron oxide, is observed.

Finally, if polarization curves are to be used, far more than three are needed to characterize the pitting potential. The pitting potential is a distributed parameter and thus ca. 10 curves are needed to

characterize the distribution. The statement in the introduction that the pitting potential can be "up to 4 times higher" (line 61) is not an appropriate description. Pitting potentials are always relative to a RE, and can only be compared as a difference. If SHE or even Li were used as a reference the difference would be the same, but the "factor" would be widely different.

- Is there enough detail provided in the methods for the work to be reproduced?

Yes

Robert G. Kelly

RESPONSE TO REVIEWERS' COMMENTS

Reviewer #1:

Nature Comm Review report: Critical role of slags in pitting corrosion of laser powder bed fusion additively manufactured 316L stainless steel in salt water

This manuscript reports a study on the origin of pitting corrosion on as-built surfaces of LPBF 316LSS. The main finding is “that Mn-rich silicate slags are responsible for pitting of the as-built LPBF material in sodium chloride due to their introduction of deleterious defects such as cracks or surface oxide heterogeneities”. It is claimed “that LPBF changes surface oxides due to rapid solidification and high temperature oxidation, leading to fundamentally different pitting corrosion mechanisms”.

Although this is an interesting and welcome study on the localised corrosion behaviour of LPBF 316LSS, the reviewer thinks that care needs to be taken in several aspects of the work:

We would like to thank the reviewer for taking the time to read our manuscript and providing insightful comments. This had a very positive impact on the quality of our manuscript. Thank you!

1. The identification of pitting corrosion of as-built SS316L surfaces was done on as-built surfaces after polarization testing in NaCl solution. It is not surprising to see anodic dissolution occurred on defects such as cracks or surface oxide heterogeneities in slags because these defects provide the lowest resistant path for polarisation currents. This is the same as polarising a metal surface with porous organic coatings – polarisation currents would certainly pass through pores in the organic coating because of much lower resistivity. In practical/natural corrosion condition under open circuit potential (OCP), however, the actual pitting initiation may be different. Although these defects in slags would still provide easy paths for electrolyte transportation, they may not be the most critical factor affecting pitting, and the actual pitting processes and mechanisms may be different. The reviewer thinks that there is need for corrosion tests at OCP in NaCl and/or FeCl₃ solutions. These tests may take a lot longer to do, but they will reveal pitting processes and mechanisms under practical corrosion conditions.

We are thankful for this comment. This is a very good point, and we took care to address it. We have conducted OCP tests in 0.6M NaCl on both top and side surfaces and added the results to the manuscript. This led to 4 new figures: Figures 3 in the main manuscript, and Figures S2, S3 and S4 in the supplementary information. We are happy to report that after OCP, representative of normal conditions, pit nucleation sites remain the slags, confirming our observations under polarization currents. This is the subject of the new paragraph on page 4 of the updated manuscript.

2. On the other hand, since the pitting testing was on the as-built surfaces after polarization testing, not under OCP, care needs to be taken when claiming the finding of ‘fundamentally different pitting corrosion mechanisms’.

As discussed in the response to the previous comment, we agree and the OCP tests we conducted based on the reviewer’s advice confirmed our claim. Not only did the reviewer’s comment allowed us to confirm the origin of pitting on as-built surfaces of the LPBF 316L SS, but it also allowed us to better capture differences between top and side surfaces, as discussed in the new paragraph page 4. Thank you!

The reviewer strongly suggests that the authors should perform pitting corrosion tests at OCP in order to ensure their key findings apply to natural pitting processes.

Nevertheless, I reviewer thinks that this is an interesting paper and worth considering publication in Nature Communications

We thank the reviewer for the insightful comments and encouraging feedback.

Reviewer #2:

This manuscript provides an extensive characterization of two different types of slag that become trapped in LPBF 316LSS. The TEM work of the authors is exemplary, and their analyses of the chemical compositions of the slags and surface films on the 316L provide important data for understanding the as-printed surface condition of AM components. The CFD work is outside my area of expertise, so I will focus on the interpretation with regards to pitting corrosion. Below I have followed the questions provided in the request for review

We would like to deeply thank the reviewer for the careful review and outstanding advice. This has undeniably improved the manuscript. Thank you!

- What are the noteworthy results?

1. Identification and characterization of the two types of slag along with strong evidence supporting different origins.
2. This may reveal my ignorance of AM processing of SS316L, but I was surprised by the presence of zinc and aluminum oxides on the as-printed surfaces at and, in some cases, near the slag. I could not find what the authors believe to be the origin of these impurities, but their presence is undeniable.
3. The correlation between Mn- and Si-bearing slags and corrosion, if substantiated, would be an important guide to powder composition development, as pointed out by the authors.

Thank you for the positive feedback!

- Will the work be of significance to the field and related fields? How does it compare to the established literature? If the work is not original, please provide relevant references.

The TEM work, especially the compositional analyses, of the slags is important. I strongly agree with the authors that as-printed surfaces are generally ignored in studies of AM corrosion, although the design versatility of AM will certainly lead to such surfaces that are inaccessible for post-build surface finishing (e.g., inside tubular structures). In addition, the observation of the formation of what seems to be a tenacious aluminum oxide on the surface could guide alloy development by purposely including it in the alloy powder.

We thank the reviewer for the positive comments on our work!

- Does the work support the conclusions and claims, or is additional evidence needed?

The work is consistent with the claims, but does not prove them. To do so would require capturing the pitting at a much earlier stage than shown in Figure 2 in which the pits are far too numerous and large to attribute initiation to the presence of the Type I slag.

We understand the reviewer's concern. Following both reviewers' comment about the need of validating that Type I slags act as pit nucleation sites at lower potential and without polarization, we performed OCP tests. This was an outstanding advice, as it allowed us to capture pits inside the slags before they become

too large. This is shown in Figures 3 in the main manuscript, and Figures S2, S3 and S4 in the supplementary information, and discussed page 4 of the main manuscript.

- Are there any flaws in the data analysis, interpretation and conclusions? - Do these prohibit publication or require revision?

The argument that cracks in the slag oxide are what lead to preferential pitting is questionable. While Figures 3, 5, S4, and S6 do show cracks in the oxide that reach the SS surface, but the images in Figure 2 and 4 do not convince me that the pits start at the slags. The level of damage makes it impossible to prove the initiation point.

Please, see our response to the previous comment. The OCP test allowed us to verify that pits nucleated inside slags.

The fact that the pits grow UNDER the slag is not necessarily odd; the slag acts like a crevice former. In the case of Type II slags, the alumina layer formed may well protect that area of the surface.

We agree with the reviewer, this is how we discussed it in the manuscript.

Maybe I missed it, but as the slag are created during deposition, should there be stringers of them throughout the part as each layer would have its own slag? Or is remelting that occurs upon deposition of subsequent layers sufficient to completely dissolve the slags? Wouldn't some of these slags be surrounded by molten metal, but not be melted? If so, polished surfaces would ALSO show pitting at the slags, in the authors' hypothesis is correct.

This is an interesting point. We focus this study on as-built surfaces, but following the reviewer comment, we investigated the bulk material. SiO₂ melting point is about 300°C higher than stainless steel 316L but it is hard to know if slags, that also contain Mn (with a melting temperature slightly lower than 316LSS) would have the same melting temperature. However, temperatures reached inside the melt pool are way above both material melting points. The reviewer is correct that, at a time, slags could be solid and surrounded by molten steel, but they will quickly melt too. We performed SEM/EDS on cross-sections to verify whether slags are present inside the bulk material and found that slags are in vast majority at the surface. Only in a couple instances have we found a slag inside the bulk, away from the surfaces. Please see the new figure S14. Given the high melt pool temperatures and the near absence of slags inside the bulk material, we conclude that slags are remelted every layer, and carried upward and sideways as more layers are added. We added a discussion Page 9 to cover this topic.

The authors do not mention what look like intergranular attack in Figure 5c. That would also be an important observation for a material that was tested as-printed without any sensitization heat treatment.

Thank you for mentioning the lack of explanation. What seems like intergranular attacks is actually a surface pattern, associated with grain boundaries, that developed during the process as it can also be observed before the corrosion test is conducted. Please see the new Figure S12 in the supplementary information and the new discussion at the bottom of page 7.

- Is the methodology sound? Does the work meet the expected standards in your field?

The approach used to create the corrosion is not optimal. Polarization scans in 0.1M NaCl may be a reasonable first step to characterizing corrosion behavior, but they are grossly insufficient as a means to represent the corrosion conditions to which any stainless steel component will be exposed. The concentration of chloride is too low, and this leads to the need for very high levels of polarization to initiate

and grow pits ($> +0.5V(SCE)$) in all but one case shown in Figure S2). There is no natural or service environment in which such polarization can occur.

We apologize as there was a typo throughout the text. We tested all samples, both during polarization and OCP, using 0.6M NaCl as stated in the method, not 0.1M NaCl. We fixed this issue and adjusted the title of the paper to remove “salt water”, that is associated with lower NaCl concentrations. Our testing conditions are indeed representative of service environment. Again, sorry for the mistake and thank you very much for pointing this out!

The problem this level of polarization creates is in the morphology of the damage. At high polarizations, the high Cr in the cell walls is preferentially dissolved via transpassive dissolution, leading to the damage morphology shown in Figures 4c, for example. At lower potentials far more relevant to service environments (e.g., $< 0V SCE$), the interiors of the cells are preferentially attacked.

This is a very good point. We added a discussion at the bottom of page 6 to mention the reason for cell boundary preferential dissolution and referred to the paper of Prof. Kacher that explains this phenomenon.

The use of high polarization has been the bane of the corrosion field at times, and propagating it into the AM field would be unwise. This transpassive dissolution also applies to Figures 6 and 7 in which the disappearance of the Cr oxide, but retention of iron oxide, is observed.

Thank you for commenting on the challenges associated with high polarization. This is very useful for the discussion. We conducted TEM analysis on the surface of the samples tested at OCP. Please see the new Figure 8. After OCP, the Cr oxide starts to dissolve too, but not completely, and the MnSiO layer remains intact, like observed after polarization. We commented on that matter on Page 8.

The fact high polarization should not be used or used carefully to study metals corrosion is an interesting comment. This may reflect on all literature on corrosion of AM 316LSS in NaCl because polarization testing is the primary method used to study corrosion. Given the high breakdown potential attained with this material, the effect on the reported corrosion mechanisms remains to be determined and may impact the relevance of this literature body to practical use of 316LSS in seawater. We thank the reviewer for pointing us in the right direction for this study and the ones to come.

Finally, if polarization curves are to be used, far more than three are needed to characterize the pitting potential. The pitting potential is a distributed parameter and thus ca. 10 curves are needed to characterize the distribution. The statement in the introduction that the pitting potential can be “up to 4 times higher” (line 61) is not an appropriate description. Pitting potentials are always relative to a RE, and can only be compared as a difference. If SHE or even Li were used as a reference the difference would be the same, but the “factor” would be widely different.

Thank you for raising this point. We modified the statement about the pitting potential that can be “up to 4 times higher” on page 3. In addition, we repeated polarization tests to have 10 for each surface orientation, following the reviewer’s advice. It allowed us to be more statistical when discussing the differences between surfaces, together with the results from the OCP measurements. This has led to the new figure S2 and the improved discussion page 4. Thank you for making our paper stronger!

- Is there enough detail provided in the methods for the work to be reproduced?

Yes

REVIEWERS' COMMENTS

Reviewer #1 (Remarks to the Author):

The authors have addressed the main questions and concerns in my previous review report by performing new corrosion experiments at OCP.

Reviewer #2 (Remarks to the Author):

I appreciate the responses to the review. They were complete and the extra work involved in answering some of the questions raised is greatly appreciated and very impressive.